# Learned Block Iterative Shrinkage Thresholding Algorithm for Photothermal Super Resolution Imaging

**DOI:** 10.3390/s22155533

**Published:** 2022-07-25

**Authors:** Jan Christian Hauffen, Linh Kästner, Samim Ahmadi, Peter Jung, Giuseppe Caire, Mathias Ziegler

**Affiliations:** 1Communication and Information Theory, Berlin Institute of Technology, 10623 Berlin, Germany; caire@tu-berlin.de; 2Industry Grade Networks and Clouds, Faculty of Electrical Engineering, and Computer Science, Berlin Institute of Technology, 10623 Berlin, Germany; d.kaestner@tu-berlin.de; 3Department of Non-Destructive Testing, Bundesanstalt für Materialforschung und -Prüfung, 12489 Berlin, Germany; samim_193@hotmail.de (S.A.); mathias.ziegler@bam.de (M.Z.)

**Keywords:** active thermal imaging, block-sparsity, deep unfolding, defect reconstruction, iterative shrinkage thresholding algorithm, laser thermography, learned optimization, neural network, regularization

## Abstract

Block-sparse regularization is already well known in active thermal imaging and is used for multiple-measurement-based inverse problems. The main bottleneck of this method is the choice of regularization parameters which differs for each experiment. We show the benefits of using a learned block iterative shrinkage thresholding algorithm (LBISTA) that is able to learn the choice of regularization parameters, without the need to manually select them. In addition, LBISTA enables the determination of a suitable weight matrix to solve the underlying inverse problem. Therefore, in this paper we present LBISTA and compare it with state-of-the-art block iterative shrinkage thresholding using synthetically generated and experimental test data from active thermography for defect reconstruction. Our results show that the use of the learned block-sparse optimization approach provides smaller normalized mean square errors for a small fixed number of iterations. Thus, this allows us to improve the convergence speed and only needs a few iterations to generate accurate defect reconstruction in photothermal super-resolution imaging.

## 1. Introduction

Active thermal imaging is a nondestructive testing technique applied in many areas such as in production industries (3D printing, car/aerospace manufacturing [1,2,3]), in medicine (breast cancer diagnosis [4,5,6]), or in dentistry [7]. Most of the research in active thermal imaging deals with the location of anomalies or defects in materials. These defects are detected by actively heating up a specimen, e.g., by laser or flash illumination, which yields photothermal imaging [8]. The resulting heat diffusion after heat excitation is observed with an infrared (IR) camera. The observed heat flow by the IR camera is then evaluated using the generated thermal film sequence. Defect indications are clearly visible in these thermal film sequences if the heat accumulates at a void or if changes in the heat flow gradient are visible due to a crack. Instead of photothermal imaging, other thermographic imaging techniques can be applied, where e.g., induction coils are used as a heating source [9]. Apart from active thermography, there are other nondestructive testing techniques such as ultrasound testing (UT), x-ray computed tomography (CT), and eddy current testing (ET). Active thermal imaging outperforms these techniques as it is a contactless technique (UT needs contact), is relatively inexpensive and flexible in terms of specimen sizes (compared to CT), and also enables us to investigate deeper-lying defects (compared to ET). However, active thermography suffers from low spatial resolution due to the blurring nature of heat diffusion. Therefore, it is of huge interest to overcome the spatial resolution limitations, which can be circumvented by utilizing suitable image processing techniques.

Knowing the resulting heat distribution by active thermal imaging for a defect-free body, the exact defect distribution of the defective body can be found by solving an underlying inverse problem [10]. This severely ill-posed thermal inverse problem can be solved by common techniques such as singular value decomposition or Bayesian approaches [11,12]. However, these techniques are computationally intensive in contrast to sparsity-exploiting techniques such as the least absolute shrinkage and selection operator (LASSO) methods [13].

Sparse signal reconstruction is becoming increasingly popular, especially in the industrial sector. These algorithms are highly attractive as, in the field of compressed sensing, knowledge of the device or target can be exploited for a high-quality reconstruction [14,15]. Consequently, in the last few years, optimization algorithms like ISTA, FISTA [16], learned algorithms such as LISTA [17], and adaptions like ALISTA [18] or LISTA-AT [19] gained much attention, especially in the field of computational imaging. Unfolding methods are already established in research and are used to solve inverse problems [20]. Especially for multiple-measurement-vector (MMV) problems, group LASSO as a block-regularization method can be applied to find an accurate solution [21,22]. Block-sparse recovery has been extensively studied with performance guarantees, also in fields of applied mathematics [23,24,25] and has been recently applied to active thermal imaging for defect detection in nondestructive testing [26,27,28,29,30,31,32]. Using multiple and different blind illumination patterns for heating in active thermal imaging results in an MMV problem. Blind illumination patterns assume that the exact position of illumination is unknown in industrial applications [29]. This occurs, for example, when the heat source or the specimen is held by a robot that has a certain amount of positional noise. Unfortunately, so far, one needs to carefully choose the regularization parameters for the block optimization problem due to the sensitivity of the thresholding [31], which made this approach tedious to implement. In addition, it is very likely that an optimal manual choice of these parameters differs for different investigated specimens. Additionally, it is unclear whether the optimal parameters have been really found. Finally, the algorithm needs hundreds of iterations to reach convergence even if the optimal regularization parameters have been found.

These algorithms are analytically well understood, but difficult to apply in practice. The main disadvantage of these classical approaches to solving the inverse thermal problem is the high computational effort, e.g., due to many iterations or expensive computations. Additionally, these algorithms depend on parameters, such as the block sparse regularizer, which directly affects the reconstruction quality or stability of the respective method. They are therefore impractical, especially in the manufacturing industry where one wants to reconstruct defects quickly and accurately to see if a large number of samples are defective. In this work, a solution is provided for the aforementioned problem by utilizing a learned iterative joint sparsity approach called the learned block iterative shrinkage thresholding algorithm (LBISTA). This approach applies neural network learning, e.g., a suitable choice of the regularization parameters or the training weights. In our approach, the training data are generated synthetically using uniformly random distributed defect distribution and corresponding thermal film sequences. Utilizing LBISTA for photothermal super resolution (SR) imaging avoids the manual choice of regularization parameters and allows for a higher convergence speed than the block iterative shrinkage thresholding algorithm (Block-ISTA or BISTA). It should be noted that LBISTA needs training times of a few hours, depending on the used hardware, to establish a trained network. Once the network is trained, LBISTA can be applied to photothermal SR imaging and generates super-resolved images much faster than BISTA. Thus, this could be highly attractive in production industries such as in 3D printing, where thermal imaging is used to monitor the processing [33,34,35,36], avoiding further post-inspections after the 3D printing has finished [26] and thus enables smaller processing time and costs. In general, such a learned block-sparse recovery approach is not new as learned group LASSO approaches already exist [37,38,39]. In [40] we were able to apply this idea to various BISTA-type algorithms but we did not focus on the theoretical aspects of this approach. Thus, the novel and main contributions of this manuscript are:implementation of LBISTA as a method to solve the inverse problem in photothermal SR imaging for nondestructive testing without a manual choice of regularization parameters;results of applying LBISTA to synthetic and experimental test data based on experiments performed with a specimen made of steel S235JR to examine defects that are not resolvable with conventional flash thermography;comparison of LBISTA with state-of-the-art BISTA for photothermal SR imaging;parameter studies of (i) tied and (ii) untied LBISTA: (1) studies of the hyperparameters, (2) studies of the parameters set to generate the synthetic training data (forward model) for photothermal SR imaging. In the tied case we train the same particular trainable parameter for each layer and in the untied case we can train it for each layer individually.

The outline of this manuscript is as follows. Section 2 explains the mathematical model in active thermal imaging and in photothermal SR imaging. In this section, it is shown how photothermal SR imaging can be cast into a block-sparse recovery problem. In Section 3, the implementation of LBISTA is shown, i.e., the description of the training data and the description of the proposed algorithms (the code is included in https://github.com/BAMresearch/Photothermal_SR_Net, accessed on 23 January 2020). Section 4 presents the results after applying LBISTA to synthetic data and to experimental data acquired from thermal imaging during structured laser illumination measurements. The results are compared with state-of-the-art BISTA in conventional photothermal SR imaging (where Block-ISTA or Block-FISTA are often used as optimization techniques within the iterative joint sparsity approach, see e.g., [32]). In addition, parameter studies are shown indicating the influence of the hyperparameters and the training parameters set to generate synthetic training data on the performance of LBISTA. Finally, in Section 5, the main achievements with LBISTA are listed and some future perspectives are given.

## 2. Mathematical Model in Active Thermal Imaging

In these studies, we work with a defect pattern assuming it does not change over the height (lines as defects, see Figure 1) so that we can simplify our model by calculating the mean over the vertically arranged pixels (see dimension y in Figure 1). The IR camera measures temperature values that can be described in active thermal imaging by a convolution in space and time (denoted in the following as ${*}_{r,t}$) of the fundamental solution of the heat diffusion equation and the absorbed excitation energy (see Green’s function solution approach in [10]). In the following, these measured temperature values are described by a discrete temperature matrix
(1)$$\begin{array}{c} T=\phi_{\mathrm{PSF}}{*}_{r,t}X \end{array}$$
with $T\in R^{N_{r}\times N_{t}}$. $\phi_{\mathrm{PSF}}\in R^{N_{r}\times N_{t}}$ represents the discrete equivalent of the fundamental solution of the heat diffusion equation, where $N_{r}$ stands for the number of measured pixels in the dimension *r* and $N_{t}$ for the number of measured images in time domain. PSF indicates that the values in the matrix refer to the well-known thermal point spread function (PSF). The discrete equivalent of the absorbed heat flux density is designated as $X\in R^{N_{r}\times N_{t}}$. It considers the irradiance in discrete space (*r*) and time (*t*) dimension denoted by $I_{r,t}\in R^{N_{r}\times N_{t}}$ as well as the absorption coefficient of the material under investigation in discrete space denoted by $a\in R^{N_{r}}$, so that $X=I_{r,t}\circ A$, where ∘ is the Hadamard-product and $A=\left[a\ldots a\right]\in R^{N_{r}\times N_{t}}$

As described in the introduction, we are dealing with blind structured illumination, which means that we do not know the exact illuminated positions at the specimen under investigation using a laser as a heat source. This results in a small but relevant change of Equation (Equation 1) into:(2)$$\begin{array}{c} T^{m}=\phi_{\mathrm{PSF}}{*}_{r,t}X^{m}=\phi_{\mathrm{PSF}}{*}_{r,t}\left(I_{r,t}^{m}\circ A\right), \end{array}$$
with $m=1\ldots N_{\mathrm{meas}}$, where $N_{\mathrm{meas}}$ denotes the number of measurements. The spatial and temporal distribution of the absorbed heat flux density $X^{m}=I_{r,t}^{m}\circ A$ varies for each measurement *m* as the illuminated spatial position varies for each measurement *m*. This represents an MMV problem in the forward problem formulation.

### 2.1. Defect Detection and Reconstruction

In our case, defects on a material are physically described as a change of optical absorption coefficient since the defect region (high value: max{a}∼0.95) differs from the defect-free region (low value: min{a}∼0.15) in its material properties. Thus, the absorption pattern matrix A (see Equation (Equation 2)) represents the defect pattern that is of interest. Since we assume blind structured illumination, we cannot separate the illumination $I_{r,t}^{m}$ from the absorption pattern matrix A. However, we can reformulate Equation (Equation 2) by extracting the known illumination pulse duration $i_{t}\in R^{N_{t}}$, which does not change over the measurements, from $I_{r,t}^{m}=i_{r}^{m}⊗i_{t}$ with $i_{r}^{m}\in R^{N_{r}}$ yielding:(3)$$\begin{array}{c} \begin{array}{cc} T^{m} & =\phi_{\mathrm{PSF}}{*}_{r,t}\left[\left(i_{r}^{m}⊗i_{t}\right)\circ A\right]\\ \; & =\left(\phi_{\mathrm{PSF}}{*}_{t}i_{t}\right){*}_{r}\left[\left(i_{r}^{m}⊗1\right)\circ A\right]\\ \; & =\phi {*}_{r}X_{r}^{m}, \end{array} \end{array}$$
where $\phi \in R^{N_{r}\times N_{t}}$ represents the thermal PSF, that considers the illumination pulse length. Moreover, $X_{r}^{m}\in R^{N_{r}\times N_{t}}$.

### 2.2. Photothermal Super Resolution

The photothermal SR technique refers to structured illumination in space and promotes high resolvability by e.g., illuminating a specimen using a narrow laser line. The realization of photothermal SR is possible if a lot of measurements/illuminations are performed with the narrow laser line resulting in an MMV problem. In our previous studies, we have shown that one can scan step by step with a single laser line with submillimeter position shifts to provide high resolvability [31]. This technique relies on spatial frequency mixing of the illumination pattern and the target pattern, here the absorption pattern. Spatial frequency mixing allows us to generate higher frequency components, enabling super-resolution. Since $i_{r}^{m}$ has a certain width in space according to the used narrow laser line width, the use of photothermal SR results in equal spatial distributions for $\sum_{m}i_{r}^{m}$ and a (see [31]). Hence, the initial goal is to determine the variable $X_{r}^{m}$. We measure $T^{m}$ with the IR camera and we can determine ϕ analytically. This underlying inverse problem has been considered many times in our studies [29,30,31,32] and a promising approach to obtain $X_{r}$ was block-regularization.

### 2.3. Block-Minimization Problem

Before using block-regularization, we eliminate the time dimension to reduce the data size. The straightforward way shown in our previous studies [30,31,32] is to extract one thermogram per measurement in the time domain, which exhibits the highest SNR—this is the so-called maximum thermogram (MT) method. We can reformulate Equation (Equation 3) after applying the MT method separately to each measurement *m* yielding the reduced (reduc) data with:(4)$$\begin{array}{c} T_{\mathrm{reduc}}^{m}=\phi_{\mathrm{reduc}}{*}_{r}x_{\mathrm{reduc}}^{m} \end{array}$$
with $T_{\mathrm{reduc}}^{m}$, $\phi_{\mathrm{reduc}}$, and $x_{\mathrm{reduc}}^{m}\in R^{N_{r}}$. To cast this problem into a block-sparse problem, we use the knowledge that the absorption pattern remains the same for each measurement, i.e., for each $m=1,\ldots ,N_{meas}$ we have $\mathrm{supp}\left(x_{\mathrm{reduc}}^{m}\right)\subset \mathrm{supp}\left(a\right),$ where
$$\begin{array}{c} \mathrm{supp}\left(x_{\mathrm{reduc}}^{m}\right)=\left\{k\;|\;|x_{\mathrm{reduc}}^{m}\left[k\right]|\neq 0\;,\;k=1,\ldots ,N_{r}\right\}. \end{array}$$

By casting these measurements into a matrix
(5)$$\begin{array}{cc} X_{\mathrm{reduc}}= & \left(x_{\mathrm{reduc}}^{1},\cdots ,x_{\mathrm{reduc}}^{m},\cdots ,x_{\mathrm{reduc}}^{N_{meas}}\right)\in R^{N_{r}\times N_{meas}} \end{array}$$
we obtain the block-sparse vector $\tilde{x}=\mathrm{vec}\left(X_{\mathrm{reduc}}^{T}\right)\in R^{N_{r}\cdot N_{meas}}$ with $N_{r}$ blocks of length $N_{meas}$. Using e.g., Block-ISTA then tries to minimize the following term:(6)$$\begin{array}{cc} {\hat{X}}_{\mathrm{reduc}}= & \arg\min_{X_{\mathrm{reduc}}}\sum_{m=1}^{N_{\mathrm{meas}}}\sum_{k=1}^{N_{r}}|\left(\phi_{\mathrm{reduc}}{*}_{r}x_{\mathrm{reduc}}^{m}\right)\left[k\right]-T_{\mathrm{reduc}}^{m}\left[k\right]{|}^{2}\\ & +\lambda ∥X_{\mathrm{reduc}}{∥}_{2,1} \end{array}$$
with the block- or joint-sparsity inducing $\ell_{2,1}$-norm $∥X_{\mathrm{reduc}}{∥}_{2,1}=\sum_{k=1}^{N_{r}}\sqrt{\sum_{m=1}^{N_{\mathrm{meas}}}{\left|x_{\mathrm{reduc}}^{m}\left[k\right]\right|}^{2}}$ this is important as we are working with blind structured illumination. Block-ISTA is stated as the following fixed-point iteration (inspired by [27,41]) and were also used in our previous studies [30,31,32] to solve (Equation 6),
(7)$$\begin{array}{c} {\hat{x}}_{\left(i\right)}^{m}=\eta_{\lambda }\left({\hat{x}}_{\left(i-1\right)}^{m}-2\gamma \phi_{\mathrm{reduc}}{*}_{r}\left(\phi_{\mathrm{reduc}}{*}_{r}{\hat{x}}_{\left(i-1\right)}^{m}-T_{\mathrm{reduc}}^{m}\right)\right), \end{array}$$
$i=1,\ldots ,N_{\mathrm{iter}}$. The number of iterations $N_{\mathrm{iter}}$ is chosen such that convergence is reached and the soft block-thresholding operator $\eta_{\lambda }$ can be computed by:(8)$$\begin{array}{c} \eta_{\lambda }\left({\hat{x}}^{m}\right)\left[k\right]=\max\left\{0,1-\frac{\lambda }{\sqrt{\sum_{m=1}^{N_{\mathrm{meas}}}{\left|{\hat{x}}^{m}\left[k\right]\right|}^{2}}}\right\}{\hat{x}}^{m}\left[k\right]. \end{array}$$

The step size is defined by $\gamma \in (0,\frac{1}{L}]$, where *L* is the Lipschitz constant of the gradient of the data fitting term. The value for *L* can be determined empirically or by $L=2\pi ∥{\hat{\phi }}_{\mathrm{reduc}}{∥}_{\infty }$ where ${∥\cdot ∥}_{\infty }$ denotes the supremum/infinity norm and ${\hat{\phi }}_{\mathrm{reduc}}$ denotes the Fourier transform of $\phi_{\mathrm{reduc}}$[27]. The value for λ is chosen empirically for (Equation 7)—Block-ISTA, e.g., $\lambda =4\times 10^{-3}$. The same value was used in our previous work, where we utilized Block-FISTA to examine the same specimen [30].

## 3. Learned Block Iterative Shrinkage Thresholding Algorithm

The success of LBISTA is strongly dependent on the choice of the training data and on the implementation of the training which will be explained in detail in the following subsections.

### 3.1. Training Data

To create the training data, we follow the forward problem shown in Equation (Equation 4). To define $x_{\mathrm{reduc}}^{m}$, we have different parameters, which can be varied: the defect width (see the width of a black stripe in Figure 1), the laser line width (see the width of a single laser line shown in Figure 1), the absorption coefficient for a defective and a defect-free region, the number of defects (probability of nonzero in space for a if we assume that the defect-free regions have an absorption coefficient of 0, i.e., the element $a_{i}$ of a is equal to 1 with a given probability indicating a defect - we then widen these defects according to the defect width, i.e., setting the respective elements of a also to 1), and the number of laser lines (probability of nonzero in space for $i_{r}^{m}$ if we assume that $i_{r}^{m}$ equals zero at the nonilluminated positions, works similar to generating a). Hence, we implemented $x_{\mathrm{reduc}}^{m}$ as $i_{r}^{m}\circ a$, determined $\phi_{\mathrm{reduc}}$ analytically and calculated $T_{\mathrm{reduc}}^{m}$.

### 3.2. Training Implementation

Inspired by the code implementation of [42] we consider layer-wise learning instead of end-to-end learning for LBISTA. Each network depends on a set Θ of trainable variables and on a number of layers *K*. We define the layers for tied LBISTA as follows
(9)$$\begin{array}{c} {\hat{x}}_{\left(i\right)}^{m}=\eta_{\lambda^{\left(i-1\right)}}\left(s*{\hat{x}}_{\left(i-1\right)}^{m}+b*y^{m}\right) \end{array}$$
and for untied LBISTA
(10)$$\begin{array}{c} {\hat{x}}_{\left(i\right)}^{m}=\eta_{\lambda^{\left(i-1\right)}}\left(s^{\left(i-1\right)}*{\hat{x}}_{\left(i-1\right)}^{m}+b^{\left(i-1\right)}*y^{m}\right), \end{array}$$
$m=1,\ldots ,N_{\mathrm{meas}}$. Note that we simplified the fixed point representation with $*:={*}_{r}$ and $y^{m}:=T_{\mathrm{reduc}}^{m}$. We define the set of trainable variables $\Theta_{tied}=\left\{s,b\right\}\cup {⋃}_{i=1}^{K}\Theta_{tied}^{\left(i\right)}$ for tied LBISTA (Equation 9), where
$$\begin{array}{cc} \Theta_{tied}^{\left(i\right)} & =\left\{\lambda^{\left(i-1\right)}\right\},\;i=1,\ldots ,K. \end{array}$$
We initialize these variables with $b=2\gamma \phi_{\mathrm{reduc}}\in R^{N_{r}}$, $s=e-b*\phi_{\mathrm{reduc}}\in R^{N_{r}}$, where $e={\left[1,0,\ldots ,0\right]}^{T}\in R^{N_{r}}$ for some step size $\gamma \in (0,\frac{1}{L}]$. In addition, we use the same initialization for the regularization parameter in each layer i=0,…,K−1 with $\lambda^{\left(i\right)}=4\times 10^{-3}$.

For untied LBISTA (Equation 10) we define $\Theta_{untied}={⋃}_{i=1}^{K}\Theta_{untied}^{\left(i\right)}$, where
$$\begin{array}{cc} \Theta_{untied}^{\left(i\right)} & =\left\{s^{\left(i-1\right)},b^{\left(i-1\right)},\lambda^{\left(i-1\right)}\right\},i=1,\ldots ,K. \end{array}$$

Untied means that each layer has its own set of trainable variables s,b, we use the same initialization for each layer. Therefore, we initialize $b^{\left(i\right)}=2\gamma \phi_{\mathrm{reduc}}$, $s^{\left(i\right)}=e-b*\phi_{\mathrm{reduc}}$, in each layer i=0,…,K−1. These variables are initialized such that we would obtain the original Block-ISTA if we do not apply the training.

The training follows Algorithm 1 and can take a large amount of time depending on the choice of the termination condition and therefore the number of iterations for each layer and each refinement. Hence, the training time also depends on the number of refinements and on the number of layers. The calculation time for one iteration is around 150  ms. Within a layer, we usually have 10,000 iterations and with each refinement, an additional 1000 iterations. We have used the GPU Quadro RTX 8000 to perform the training.
**Algorithm 1:** LBISTA, implementation of training
**Input :**Training rate *t_r_*, refinements *f*, exact solution **x**^*∗,m*^ and trainable Variables *V* with *case* = *tied or case* = *untied*

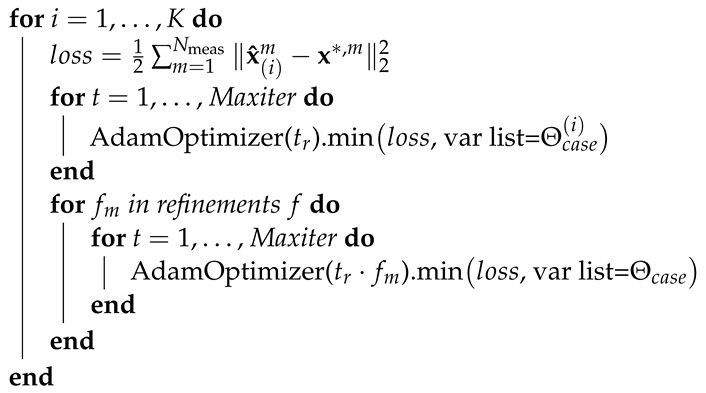



## 4. Evaluation of LBISTA

### 4.1. Numerical Results

To study the performance of the LBISTA, we first show numerical studies considering the block-sparse convolution problem $T_{\mathrm{reduc}}^{m}=\phi_{\mathrm{reduc}}{*}_{r}x_{\mathrm{reduc}}^{m}$. There is also a theoretical analysis of this approach in the setting of sparse recovering—in [18,43] a lower and upper bound for the reconstruction error is derived. This could also be applied to the block-setting and especially to the setting of photothermal SR imaging, but this is part of ongoing work.

In the following, we use the same notation as shown in Equation (Equation 4) with $y_{\mathrm{train}}^{m}=$$\phi_{\mathrm{reduc}}{*}_{r}x_{\mathrm{train}}^{m}$ and $y_{\mathrm{train}}^{m}$, $x_{\mathrm{train}}^{m}\in R^{N_{r}}$. An analytical formulation of $\phi_{\mathrm{reduc}}$ can be found in [30]. Note that $\phi_{\mathrm{reduc}}$ is generated after applying the MT method. Table 1 shows the used parameters for training with synthetic data based on $x_{\mathrm{train}}^{m}=i_{\mathrm{train}}^{m}\circ a$. More precisely, the cells corresponding to defect pattern determine the values in a and the cells corresponding to illumination pattern the values in $i_{\mathrm{train}}^{m}\in R^{N_{r}}$. Further, Gaussian noise is added to generate $y_{\mathrm{train}}^{m}$ based on the shown SNR value in the table. For the determination of Gaussian noise variance $\sigma_{\mathrm{noise}}^{2}$, we have used: SNR = $\mu^{2}/\sigma_{\mathrm{noise}}^{2}=8\;\mathrm{dB}$ with the squared signal mean $\mu^{2}=\mathrm{PNZ}$. PNZ denotes the probability of nonzeros. LBISTA is trained for the dimensions $N_{r}=1280$ (number of pixels), $N_{\mathrm{meas}}=150$ (number of measurements based on differently chosen illumination and therefore differently generated $x_{\mathrm{reduc}}^{m}$). The maximum number of iterations Maxiter has been decreased for untied LBISTA since updating more variables causes higher computation times for each iteration.

In the following, the training data are determined by $x_{i,\mathrm{train}}^{m},y_{i,\mathrm{train}}^{m}\in R^{N_{r}}$ and i∈{1,…,$\;N_{B}\}$, $m\in \{1,\;\ldots ,\;N_{\mathrm{meas}}\}$, so that ${\left(X_{i,\;\mathrm{train}}\right)}_{i=1}^{N_{B}}={\left(x_{i,\;\mathrm{train}}^{1},\ldots ,x_{i,\;\mathrm{train}}^{N_{\mathrm{meas}}}\right)}_{i=1}^{N_{B}}\in R^{N_{B}\times N_{r}\times N_{\mathrm{meas}}}$ and similar for ${\left(Y_{i,\;\mathrm{train}}\right)}_{i=1}^{N_{B}}\in R^{N_{B}\times N_{r}\times N_{\mathrm{meas}}}$, e.g., Figure 2b. LBISTA reaches a smaller normalized mean square error (NMSE) much faster than BISTA, Figure 2e,f. According to the curves in (d), LBISTA outperforms BISTA in terms of reconstruction using $\phi_{\mathrm{reduc}}$ (see (a)) and $y_{\mathrm{test}}$ (see (c)). Of course, it strongly depends on the choice of λ how well BISTA reconstructs. Therefore, we chose the same initial values of λ for BISTA and LBISTA to have a better comparison.

Figure 2 shows the results with synthetic test data $y_{\mathrm{test}}\in R^{N_{r}}$ and ground truth $x_{\mathrm{test}}\in R^{N_{r}}$. In the following section we use real measurement test data instead of $y_{\mathrm{test}}$ on LBISTA, but still trained with synthetic training data.

### 4.2. Evaluation of LBISTA with Experimental Data from Active Thermal Imaging

Instead of creating test data synthetically as in the previous section, we apply LBISTA to real experimental data, which were measured with the IR camera (InfraTec ImageIR 9300, 1280×1024 pixels full frame, spectral range: 3–5μm). We have used the same dataset as in a previous publication (see [30] as a reference). We have used the IR camera in transmission configuration (illumination on the front of the specimen and observation of the back side with the IR camera). The specimen has a thickness of 3mm and 5 blackened stripe pairs on the front, see Figure 1. All in all, we performed around 150 measurements, resulting in m=1…150, 30 measurements per pair. Within these 30 measurements, we shifted the position by 0.4mm twice: once after the first and again after the second ten measurements. After 30 measurements the position was shifted to the next pair. This is repeated until we scanned the whole specimen (Figure 1). Each measurement differs by randomly (uniform distribution) switching on a certain number of laser lines out of twelve (at least one laser line is turned on for each measurement). Figure 3 shows the experimental test data after removing the dimension *y* by calculating the mean over the vertically arranged pixels and eliminating the time dimension by applying the MT method ([30,31,32]). Nevertheless, there is a disturbance in the data at position 41mm represented by a straight line over all measurements. This comes from the fact that the specimen had a marker line on the rear side to have an orientation about the defects in thermographic transmission configuration, Figure 4.

In the following, we compare the result after applying BISTA or LBISTA. Calculating the sum over all measurements as explained in Section 2.2 results in an approximation of the defect pattern, as shown in Figure 5. The dashed blue curve indicates the outcome of the BISTA algorithm and shows only more or less good indications for the third pair with the largest distance between the stripes (around position 37…44mm). The result of block fast ISTA (Block-FISTA), the purple dashed line, also shows only a more or less good indication for the defects, but still improves the resolution and even shows the fourth defect. Block-FISTA is also a block iterative shrinkage algorithm but with a better convergence rate than BISTA [16,27], but also has the main drawback of finding optimal regularization parameters empirically. In contrast, the application of LBISTA results in very good indications for three of five pairs. Only the stripes with the largest and closest distance to each other (position 11…14mm) cannot be clearly recognized. The other three stripes are very well reconstructed, even the stripe width can be recognized. Since the marker line at position 41mm disturbs the pattern recognition, the stripes with the largest distance to each other are hard to resolve. For now, parameters from Table 1 have been used, but it is still unclear whether these are the best choice. Therefore, we varied some of these to study their impact on the result of tied and untied LBISTA, see Figure 6 and Figure 7, respectively.

Figure 6:Training batch size (a,b): Varying the number of batches between $N_{B}=100$ and $N_{B}=200$ does not really change the result.Defect pattern (c,d,e,f): In contrast, changing the sparsity of the defects fromPNZ∈[0.005,0.03] (c.f. (e,f)) or the defect width (see (c,d)) leads to significant changes in the results such that the result in (d) is even able to clearly detect both stripes in the first pair with the smallest distance between the stripes. In (f), obviously a too small sparsity has been used as the PNZ is quite high in comparison to the chosen PNZ in (e). Thus, it is obviously beneficial to know roughly the sparsity.Training rate and number of layers (g,h): In (g) we get similarly good results just by using one refinement instead of three as used in Figure 5. Only using one refinement could save a lot of time during training. In (h), a rather bad result is shown, where we used one refinement and only three layers. Thus, these parameters should be chosen high enough so that we reach convergence within training.

Figure 7:Absorption coefficient and training iterations (a,b): In contrast to tied LBISTA, untied LBISTA exhibits significant changes by varying the absorption coefficient in training. According to our studies, an absorption coefficient of {0,1} is a good choice. Further, increasing the maximum number of iterations as shown in (b) can enhance the reconstruction quality so that all stripes could be indicated very well except for the middle stripe pair (most likely due to the marker line).Training batch size (c,d): The result in (c) using fewer batches and iterations as in (b) shows that similarly good results can be achieved. Increasing the number of batches to $N_{B}=200$ as shown in (d) can further enhance the reconstruction quality as now even the middle defect pair could be clearly resolved.Defect width (e,f): The variation of the defect width rather degrades the reconstruction result as shown in (e,f). This means that the default choice performs best for untied LBISTA. However, with tied LBISTA (Figure 6c,d) we could see improvements by changing the parameter of the defect width.Defect sparsity (g,h): The variation of the sparsity in untied LBISTA in (g,h) confirms our investigations in tied LBISTA (see (e,f)), deterioration by using a too small sparsity of PNZ=0.03.

## 5. Conclusions and Outlook

The application of LBISTA leads to much more promising results than the state-of-the-art BISTA. With LBISTA, we can omit manually selected regularization parameters and obtain the results faster and with higher reconstruction qualities/smaller NMSE for a small fixed number of iterations. This paper showcases the improvements in reconstruction for different cases: (1) convolution measurements with synthetic test data (see Figure 2); (2) convolution measurements with experimental test data from photothermal measurements (see outstanding results for tied LBISTA in Figure 5 and Figure 6d and for untied LBISTA e.g., in Figure 7d). These are based on the same training dataset. In all cases, it could be observed that untied LBISTA provides more reliable results than tied LBISTA in terms of reconstruction quality. Moreover, the parameter studies encouraged us to use as many batches, iterations, and layers as possible to achieve high reconstruction qualities. Further, a very precise model of the experiment is necessary to provide accurate reconstruction results.

Thus, the application of learned regularization algorithms, such as the proposed LBISTA, is highly recommended and attractive for industrial applications where the user does not have to choose regularization parameters manually and benefits from the remarkable speed of convergence of LBISTA. LBISTA therefore enables a less complicated evaluation of the photothermal SR data and offers the possibility of reliable in situ inspections.

As an outlook, we will study other learned block-regularization techniques based on unfolding algorithms such as FISTA or Elastic-Net. In addition, we will further study how to increase the performance of the training to train with larger datasets where we do not have to eliminate the time dimension or calculate the mean over the vertically arranged pixels.

## Figures and Tables

**Figure 1 sensors-22-05533-f001:**
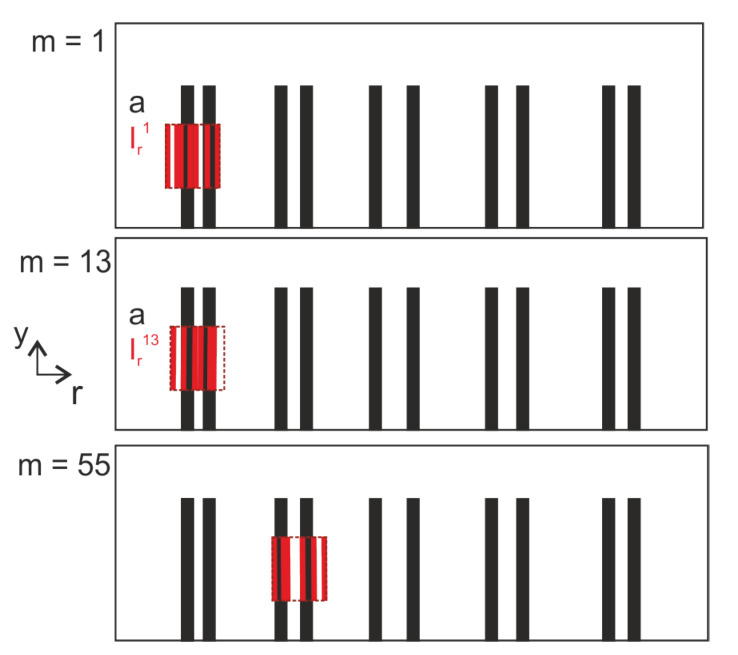
Exemplary specimen with defects shown as blackened stripes. A laser line array with twelve laser lines is used. Here *y* stands for the spatial vertical dimension and *r* stands for the spatial horizontal dimension. The illumination pattern differs for each measurement in the illuminated position and in the number of laser lines (indicated by the red area) which are switched on (randomly chosen). The dashed frame around the pattern indicates the covered illuminated area if all laser lines are switched on.

**Figure 2 sensors-22-05533-f002:**
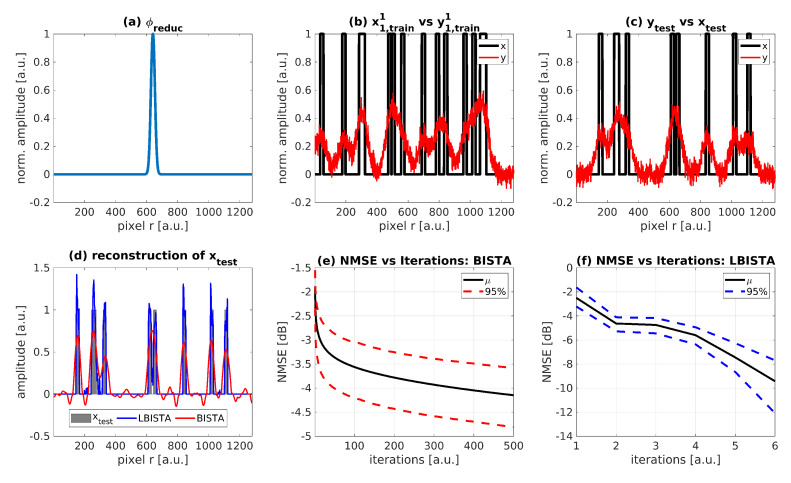
Exemplary synthetic training and synthetic test data as well as NMSE performance studies. (**a**): shape of used $\phi_{\mathrm{reduc}}$ in training, (**b**,**c**): comparison of the curve shapes for exemplary training and test datasets with e.g., $x_{1,\;\mathrm{train}}^{1}\in R^{N_{r}}$, (**d**): reconstruction of $x_{\mathrm{test}}$ using convolution-based tied LBISTA (6 iterations/layers) vs. BISTA (1000 iterations), (**e**): NMSE over iterations for BISTA with 95% confidence interval, (**f**): NMSE over iterations for tied LBISTA with 95% confidence interval.

**Figure 3 sensors-22-05533-f003:**
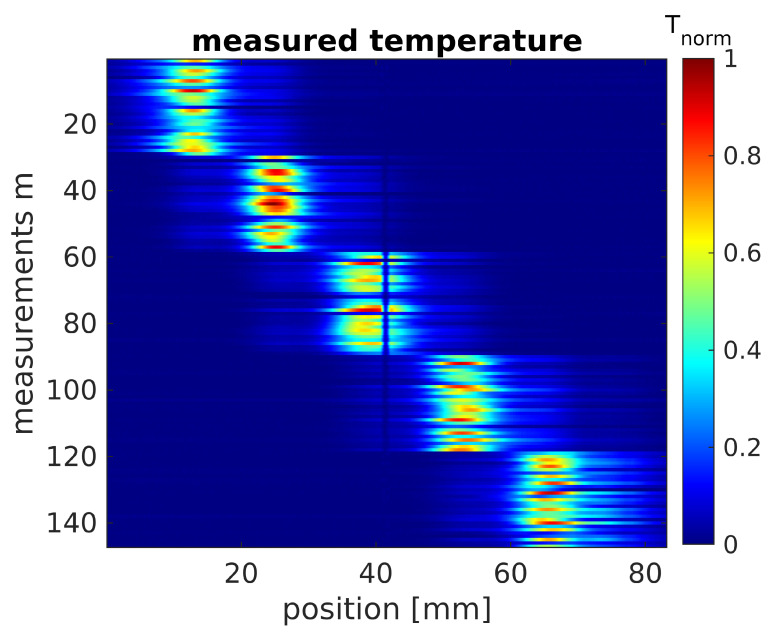
Test data from photothermal structured illumination measurements. To obtain this image, averaging over the vertically arranged pixels as well as applying the MT method (see [30,31,32]) to eliminate the time dimension is necessary. The blue marker at position = 41mm refers to the marker line on the investigated specimen (Figure 4).

**Figure 4 sensors-22-05533-f004:**
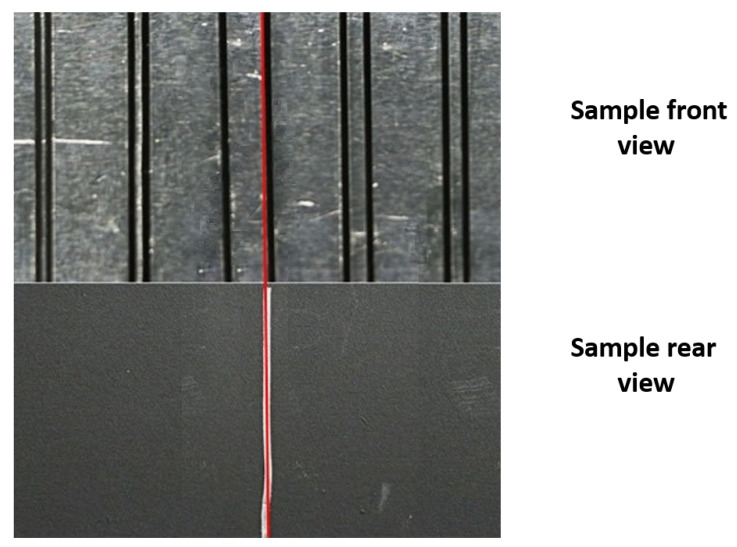
Sample front and rear view to understand origin of white marker line at position 41mm, Figure 3. The rear side of the sample has been observed by the IR camera and the front side has been excited by a laser array. The defects on the front side are shown by blackened stripes (5 pairs). The marker line on the rear side is in the same position as the left edge of the right stripe from the third pair. The distances between the pairs are (left to right): 0.5, 1, 3, 2, 1.3mm. A stripe is 1mm wide and the distance of each pair is around 10mm.

**Figure 5 sensors-22-05533-f005:**
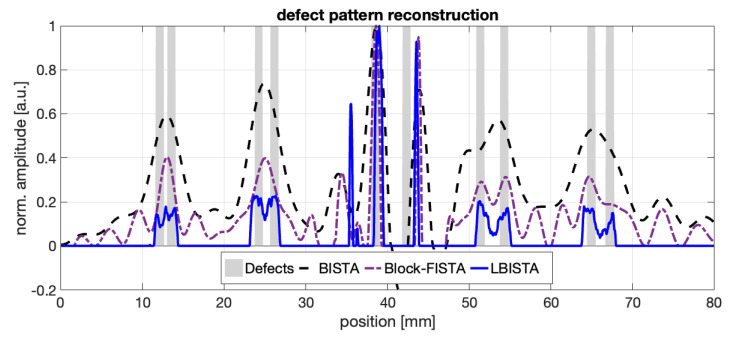
Pattern reconstruction of blackened stripes as shown in Figure 4 using BISTA, Block-FISTA, and tied LBISTA. For LBISTA the parameters in Table 1 were used. Hence, synthetic training data have been used. The BISTA and Block-FISTA result has been created by using $\gamma =\frac{1}{\sqrt{2}}$, λ=0.004, $N_{\mathrm{iter}}=500$ (selected from [30]). The marker line can be seen in the BISTA curve exhibiting normalized amplitude values <0 at position 41mm.

**Figure 6 sensors-22-05533-f006:**
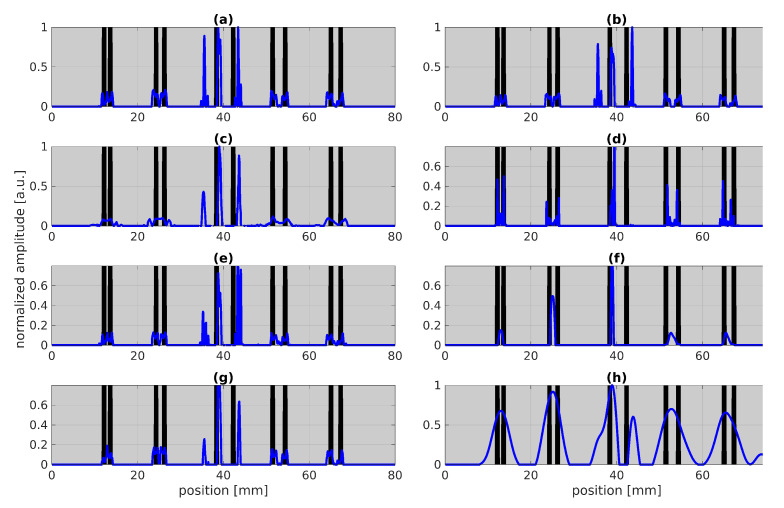
Parameter studies for tied LBISTA applied to experimental test data shown in Figure 3. The blackened areas represent the blackened stripes for better orientation. The blue curves stand for the result after applying the tied LBISTA to the experimental test data. All images are based on the parameter choice shown in Table 1. The following parameters have been changed, respectively: (**a**) $N_{B}=100$, (**b**) $N_{B}=200$, (**c**) defectwidth=0.5mm, (**d**) defectwidth=2mm, (**e**) defect sparsity PNZ=0.005, (**f**) defect sparsity PNZ=0.03, (**g**) only one refinement $f_{m}=\left\{0.5\right\}$, (**h**) only three layers K=3 and one refinement $f_{m}=\left\{0.01\right\}$.

**Figure 7 sensors-22-05533-f007:**
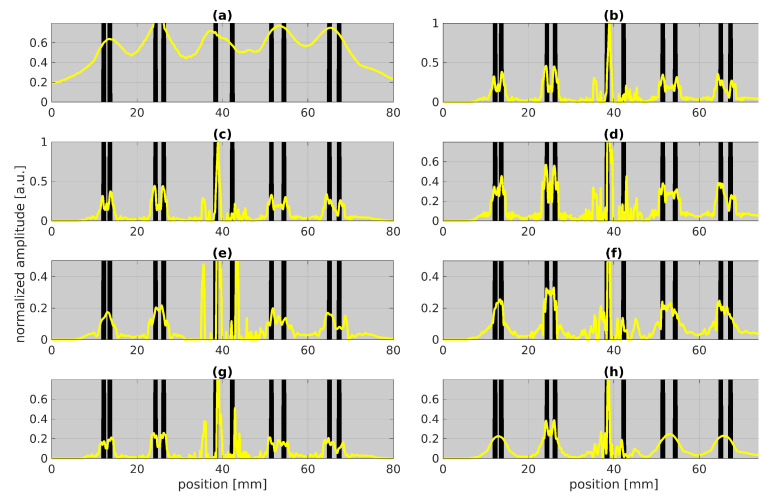
Parameter studies for untied LBISTA applied to experimental test data shown in Figure 3. The blackened areas represent the blackened stripes for better orientation. The yellow curves stand for the result after applying the untied LBISTA to the experimental test data. All images are based on the parameter choice shown in Table 1. The following parameters have been changed, respectively: (**a**) absorptioncoefficient={0.3,0.7}, (**b**) $Maxiter=10^{5}$, (**c**) $N_{B}=100$, (**d**) $N_{B}=200$, (**e**) defectwidth=0.5mm, (**f**) defectwidth=2mm, (**g**) defect sparsity PNZ=0.005, (**h**) defect sparsity PNZ=0.03.

**Table 1 sensors-22-05533-t001:** Parameters used in LBISTA for problem definition and training.

Gaussian Noise in $y_{\mathrm{train}}^{m}$	SNR = 8 dB
**Defect pattern**	defect width = 1mm
	defect sparsity (PNZ) = 0.01
	absorption coefficient = {0,1}
**Illumination pattern**	laser line width = 0.8mm
	illumination sparsity (PNZ) = 0.01
**Training parameters**	refinements *f*: $f_{m}=\left\{0.5,\;0.1,\;0.05\right\}$
	training rate $t_{r}=0.001$
	initial lambda λ=0.004
	number of layers K=6
	step size $\gamma =\frac{1}{\sqrt{2}}$
	batch number $N_{B}=150$
tied LBISTA	max. number of iterations $Maxiter=10^{5}$
untied LBISTA	max. number of iterations $Maxiter=10^{4}$

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
