# Peer review of "Learned Block Iterative Shrinkage Thresholding Algorithm for Photothermal Super Resolution Imaging"

_sensors, 2022, doi:10.3390/s22155533_

Round 1

Reviewer 1 Report

This paper proposed learned block iterative shrinkage thresholding algorithm (LBISTA) that uses synthetically generated and experimental test data from active thermography for defect reconstruction. The final results improve the convergence speed and only needs a few iterations to generate accurate defect reconstruction in photothermal super resolution imaging. The reviewer thinks this paper will be of sure interest for that audience, even if some revisions are needed to improve its impact. General comments and specific comments are given below.

1.  In the introduction part, authors should give some simple discussions to show the advantages and disadvantages of the existing results.

2.    In Fig. 1, the left column has the coordinate of y and r. Please describe it in detail.

3.   In line 138, what is ?

4.   What is LBISTA network? Please describe the architecture of LBISTA network briefly.

5.    Except BISTA, the authors should be compared with other existing works.

6.    The line241 and line255, the format of Fig.6 and Fig.7 should be the same.

Reviewer 2 Report

In this work, the authors continued their previous results by giving a further study of the demonstrated LBISTA with more details. In particular, the authors applied the algorithm to solve the inverse problem in photothermal superresolution imaging by avoiding manually regularize parameters and compared the method with conventional means such as flash thermography as well as other algorithms including BISTA. The work looks interesting, as the method also combines with neural networks learning. 

The manuscript is well organized and written. By first presenting the mathematical background underlying the LBISTA, the authors continued to implement the LBISTA and compare the results with the experimental data. The quality of the work looks reasonably high and meets the high standards of the journal Sensors.

As the work falls into the scope of the journal, I am happy to recommend it for publication. However, before its formal acceptance, I would like to welcome the authors to address the following issues in the revision:

(a)  In page 5, in line 138, "\phi\phi" should be "\phi".

(b)  In page 6, in line 157, could the authors provide more information on "a"? For example, how could one determine "a"?

(c)  In page 6, the two equations in line 167 look similar to the two equations in line 164. I guess the authors forgot the superscripts for them in line 167. Please double check that.

(d)  In page 7, in line 191, what is "N_d"? Please explain it.

(e)  In page 11, line 244 is incomplete for "PNZ".

(f)  For Figures 2-7, could the authors double check whether all parameter information is given in the text?

(g)  Many cited references have incomplete information. For examples, References 1, 11, 17, 20, 24, 25, 30, 31, 32, 33, 34, 35, 37, 38, and 43.
